# Singleton {NOT} and Doubleton {YES; NOT} Gates Act as Functionally Complete Sets in DNA-Integrated Computational Circuits

**DOI:** 10.3390/nano14070600

**Published:** 2024-03-28

**Authors:** Andrea C. Bardales, Quynh Vo, Dmitry M. Kolpashchikov

**Affiliations:** 1Chemistry Department, University of Central Florida, Orlando, FL 32816, USA; abardales@ucf.edu (A.C.B.);; 2National Center for Forensic Sciences, University of Central Florida, Orlando, FL 32816, USA; 3Burnett School of Biomedical Sciences, University of Central Florida, Orlando, FL 32816, USA

**Keywords:** Boolean logic, DNA circuits, DNA nanostructures, universal logic gates

## Abstract

A functionally complete Boolean operator is sufficient for computational circuits of arbitrary complexity. We connected YES (buffer) with NOT (inverter) and two NOT four-way junction (4J) DNA gates to obtain IMPLY and NAND Boolean functions, respectively, each of which represents a functionally complete gate. The results show a technological path towards creating a DNA computational circuit of arbitrary complexity based on singleton NOT or a combination of NOT and YES gates, which is not possible in electronic computers. We, therefore, concluded that DNA-based circuits and molecular computation may offer opportunities unforeseen in electronics.

## 1. Introduction

Boolean logic gates are the most basic components in electronic computers [1]. A set of AND, OR, and NOT gates is a well-known functionally complete set in digital computers [1]. This set has attracted attention because of its universality—the ability to achieve any other logic functions by integrating multiple units of this limited set [2]. This modular and scalable approach enables the easy design and cost-efficient manufacturing of computational circuits. IMPLY and NAND are ‘universal’ (or functionally complete) gates: each of them is sufficient to build semiconductor circuits of arbitrary complexity [3]. The IMPLY logic produces a low output only when the conditional set (Input 1: low, and Input 2: high) is true (Figure 1b). Lately, IMPLY has also attracted attention for its use in ‘memristive’ switches, memory resistors that perform logic operations [4,5]. NAND Boolean logic produces a low output only when both inputs (Input 2 and Input 3) are high (Figure 1c). The simplest Boolean logic gates are YES and NOT: YES produces a high output in the presence of the input and a low output in its absence (Figure 1a, top). NOT is the inverter of YES logic (Figure 1a, bottom). In digital computing, neither the combination of YES and NOT gates, nor NOT gates alone, have ever been reported to comprise a functionally complete set of gates.

Boolean logic gates made of small organic molecules [6], proteins [7], and nucleic acids [8] have been reported. It is believed that such gates can be used to build computational circuits that are smaller, consume less energy, and are capable of multiple parallel computing [8,9]. Furthermore, logic gates made of DNA and RNA can be used as molecular tools for diagnosis and therapy [10].

We have been developing DNA logic gates connected to each other via DNA four-way junction (4J) structures [11,12]. The gates recognize nucleic acid sequences as inputs and produce a new sequence by bringing two oligonucleotide fragments into proximity, which are the output of the 4J gates. The new output sequence can be conveniently detected by a molecular beacon (MB) probe—a fluorophore—and a quencher-labelled DNA hairpin [13]. The change in fluorescence from the opening/closing of the MB probe can be correlated to the binary response (0 and 1) as in digital computing.

Figure 2a illustrates the functional mechanism of a 4J YES Boolean logic gate with the output sequence (A_1_ + B_1_) triggering the MB_1_ probe opening after input recognition. In the 4J NOT gate (Figure 2b), strands A_2_ and B_2_ are brought together by a DNA “bridge”, which stabilizes their hybridization with the MB_2_ probe in the absence of the input, thus enabling a high fluorescent signal (digital 1). In this setting, the 4J NOT gate follows the NOT logic truth table by giving a functional output sequence (output 1) for input 0 (absence/low). The addition of an oligonucleotide input decomposes the 4J structure by hybridizing to the bridge fragment and triggering the displacement of A_2_ and B_2_, which results in the dissociation of the MB_2_ probe. This causes MB_2_ to fold itself as a hairpin and to exhibit low fluorescence (digital 0).

Here, we report that connected YES and NOT and two connected NOT gates can lead to functionally complete IMPLY or NAND gates, respectively. To facilitate communication between YES and NOT gates, we spatially localized the gates on a DNA scaffold, named here a DNA board, which is composed of four strands: Rail 1, Rail 2, Staple 1, and Staple 2 (Figure 2c). The DNA board contains a single-stranded (ss)DNA section that serves as a flexible hybridization board for the integration of multiple DNA logic units, which allows for DNA circuits to be built.

## 2. Materials and Methods

### 2.1. Materials

DNase/protease-free water was purchased from Fisher Scientific Inc. (Pittsburg, PA, USA) and used for all buffers and oligonucleotide stock solutions. MgCl_2_ (1 M solution) was purchased from Thermo Scientific (Waltham, MA, USA), 1M Tris-HCl pH 7.4 buffer from KD Medical (Columbia, MD, USA), and Triton X100 from Sigma-Aldrich (Burlington, MA, USA). All oligonucleotides were custom-made by Integrated DNA Technologies, Inc. (Coralville, IA, USA), and their stock solutions were prepared by resuspension in water and stored at −20 °C until use. The concentrations of the oligonucleotides’ stocks were determined from the Beer–Lambert equation, for which absorbance at 260 nm was measured with a Thermo Scientific Nanodrop One UV-Vis Spectrophotometer, while the corresponding extinction coefficients were determined using OligoAnalyzer 3.1 software (Integrated DNA Technologies, Inc.) (Table 1). Fluorescence assays were performed with a Perkin Elmer LS 55 Fluorescence Spectrometer (Waltham, MA, USA), Deuterium Lamp. Gel electrophoresis experiments were performed using BioRad electrophoresis equipment (Hercules, CA, USA), and visualized using BioRad Gel Doc XR+.

### 2.2. DNA Logic Gates Assembly

All DNA oligonucleotides were mixed at 200 nM in equimolar ratios in a buffer mix containing 100 mM Tris-HCl at pH 7.4, 100 mM MgCl_2_, and 0.06% Triton X100, followed by vortexing and centrifugation to make sure all the solution was dragged down. The samples were annealed by placing them in a water bath at 95 °C for 2 min and slowly cooling down to 22 °C within 8 h.

### 2.3. Fluorescence Assays

After assembly, a master mix solution was prepared containing molecular beacon (MB) probe solution and the DNA assembly. From this master mix, aliquots were dispensed in individual microcentrifuge tubes for the addition of the different inputs, followed by incubation at room temperature (22–25 °C) for 20 min. The fluorescence emission was read from those samples, containing 100 nM DNA logic gate assembly, 50 nM MB probe (12.5 nM for YES 1 and IMPLY), 100–200 nM input, 50 mM Tris-HCl at pH 7.4, 50 mM MgCl_2_, and 0.03% Triton X100.

### 2.4. Fluorescence Data Analysis

Average and standard deviations were calculated from three independent samples. To normalize the fluorescence response of each output signal, we subtracted the average fluorescence response of a MB-only solution. Each graph plots the average fluorescence difference (ΔF): fluorescence output signal − fluorescence MB signal. Error bars represent the standard deviation from three independent samples.

### 2.5. Gel Electrophoresis 

Native gels were prepared with 8% acrylamide (19:1 acrylamide/bisacrylamide) and contained 50 mM MgCl_2_. Gels were run at constant voltage (95 V) for 75 min. Samples were prepared using a 6× Cyan/Yellow loading buffer (TrackIt^TM^, Thermofisher, Waltham, MA, USA). TBE buffer (89 mM Tris Base, 89 mM boric acid, and 2 mM EDTA) was used as the running buffer. Denaturing gels were prepared to contain 8 M urea and 12% acrylamide (19:1 acrylamide/bisacrylamide). Samples were prepared using a 2× denaturing loading buffer (85% formamide, TBE, and traces of Bromophenol blue and Xylene Cyanol). Gels were run at constant voltage (150 V) and 65 °C for 1 h and 30 min. Gel-Red was used as a staining dye for the visualization of DNA bands.

### 2.6. Assembly Gel-Extraction

Next, 150 pmol of the DNA assembly was loaded into a native gel. For gel extraction, gels were run at constant voltage (100 V) and 22 °C for 1 h 30 min. The target band was identified and cut with a scalpel blade, followed by being thinly crushed, soaked in 1 mL of DNA-grade water, and incubated under shaking (120 rpm) at 37 °C for up to 24 h. The supernatant was filtered using a X-Spin Coastar filter. From the collected supernatant, DNA was precipitated by adding a 2-fold volume 2% LiClO_4_–acetone solution and separated from the supernatant by centrifugation at 10,000 RPM for 3 min (step repeated with pure acetone). The DNA pellet was dried under vacuum for 30–60 min and then resuspended with DNA-grade water.

## 3. Results and Discussion

### 3.1. IMPLY Logic Circuit (YES + NOT)

First, we optimized the performance of individual YES 1 and NOT 2 on the DNA board structure to achieve the correct digital response. Input 1 and Input 2 are the DNA sequences corresponding to has-miR-221-3p and hsa-miR-409-3p, respectively. Input 1 is recognized by YES 1, while Input 2 is recognized by NOT 2 (Table 1). Upon input recognition, YES 1 combines A_1_ and B_1_, giving an output sequence of a total of 18 nucleotides (nt) long. Conversely, NOT 2 dissociates its output sequence (17 nt) upon input recognition by Bridge strand. When only YES 1 was assembled on the DNA board, blocker strands blck A_2_ and blck B_2_ were added to cover the empty ssDNA regions on both Rail strands (Figure 3a). Similarly, when only NOT 2 was assembled on the DNA board, blck A_1_ and B_1_ were added (Figure 3c). We observed signal enhancement for YES 1 and signal reduction for NOT 2, as expected, in the presence of the input strand (Figure 3b,d).

We then integrated both YES 1 and NOT 2 gates on the DNA board such that the output of NOT 2 served as an input for YES 1, as shown in Figure 4a. In this arrangement, the system was expected to perform as a two-input IMPLY logic gate producing high output (measured as high fluorescence of the MB_1_ probe) in all input combinations except when only Input 2 complementary to NOT 2 gate was present (Figure 4d). The fluorescence assays show the correct digital response of the IMPLY gate (Figure 4c). An experimental threshold (red dash line in Figure 4c left) for the differentiation of the ON (digital 1) and OFF (digital 0) output signal of the IMPLY unit was established following the concept of the limit of detection and corresponded to the average signal of YES 1-output 0 plus three times its standard deviation (SD).

We also assessed the full assembly of the YES 1 and NOT 2 gates on the DNA board through gel electrophoresis (Figure 4b). Lane 4 shows faster mobility of the IMPLY unit than that of the DNA board alone (Lane 3). This can be explained by the higher overall negative charge of the ‘loaded’ DNA board nanostructure, which has a comparable electrodynamic volume with that of the unloaded DNA board. To prove that the major band in Lane 4 contained all the expected strands, we cut this band out of the gel, eluted its content, and analysed the content using denaturing gel electrophoresis (Figure 5). For mobility reference, individual ssDNA components were added from Lane 2 to 10. Lane 11 shows the four DNA bands corresponding to the mobility of the DNA board components: Rail 1, Rail 2, Staple 1, and Staple 2. The IMPLY full assembly was loaded to Lane 12, which shows six DNA bands corresponding to the overlapping mobility of the components of the DNA board, YES 1 (A_1_ + B_1_) and NOT 2 (A_2_ + B_2_ + bridge). The IMPLY assembly after gel extraction was loaded in Lane 13, which shows five DNA bands corresponding to the components of the DNA board, YES 1 and NOT 2′s A_2_ and bridge. B_2_ is not observed in Lane 13 (Figure 5, blue arrowhead), and since this strand is detached from the DNA board, we consider that under non-equilibrium conditions like those of gel electrophoresis, B_2_ is prone to dissociation from the major assembly and was lost from the IMPLY full assembly during gel extraction.

### 3.2. NAND Logic Circuit (NOT + NOT)

To create a universal NAND function, we loaded the DNA board with two NOT gates (NOT 2 + NOT 3). NOT 3 recognizes Input 3 (a 22 nt long ssDNA). For later connectivity with NOT 2, NOT 3 was designed to assemble in the same ssDNA region as YES 1 on the DNA board. Additionally, the NOT 3 output sequence is also recognized by MB_1_. To test the individual response of NOT 3 on the DNA board, blck A_2_ and blck B_2_ were added as replacements for NOT 2 strands (A_2_ and B_2_) to maintain the rigidity of the DNA board. NOT 3 alone showed a 3-fold reduction when Input 3 was added (Figure 6), demonstrating the digital NOT behaviour of this gate.

By connecting NOT 3 with NOT 2, we obtained a two-input NAND Boolean function, which is another functionally complete logic gate (Figure 7a). We performed similar fluorescence and gel electrophoresis assays as for the IMPLY logic unit. NAND fluorescence assays show the correct digital response as expected based on its truth table (Figure 1c and Figure 7c). Gel electrophoresis also revealed a faster mobility band corresponding to the full NAND assembly (Figure 7b, Lane 4) as compared to the unloaded DNA board (Figure 7b, Lane 3). To prove that the major band (shown by a blue arrowhead) in Lane 4 contained all NAND expected strands, we performed a similar procedure as for the IMPLY assembly, by cutting and eluting this band out of the gel and analysing its content via denaturing gel electrophoresis (Figure 8).

Denaturing gel electrophoresis (dPAGE) allows for the imaging of the individual constituents of DNA assemblies. The NAND assembly after gel extraction was loaded into Lane 13, which shows seven DNA bands corresponding to the components of the DNA board, NOT 2′s A_2_ and bridge, and NOT 3. B_2_ is not observed in Lane 13 (Figure 8, blue arrowhead) since this strand is detached from the DNA board. Therefore, in non-equilibrium conditions like those of gel electrophoresis, B_2_ is prone to dissociation from the major assembly during gel extraction; a similar result was observed for the extraction of the IMPLY assembly (Figure 5).

## 4. Discussion

One common paradigm in developing a molecular computer follows the path established by the semiconductor computer technology. This includes designing a functionally complete sets of Boolean logic gates, connecting them in circuits by integrating into a common platform, powering using (bio)chemical reactions, and achieving an easily readable signal for convenient communication with a human operator [12,14]. Applications of such computational systems in controlling gene expression and in diagnosing infectious diseases and cancer have been envisioned [15,16,17,18,19]. Thus, computers made of molecules can be explored for the application of well-developed computational living systems.

This study demonstrates that molecular (DNA) computational systems may offer opportunities unrealized in electronics. Indeed, an electronic set of YES and NOT gates has never been considered as a complete set of Boolean gates. In this work, we demonstrated for the first time that YES and NOT gates, or two NOT gates made of DNA, can be connected in a circuit that fulfils functionally complete gates, IMPLY and NAND. This was possible because the YES 1 gate in IMPLY and NOT 3 in the NAND gate recognized either the oligonucleotide input or the outputs of the upstream gates; the coexistence of these two distinct functions is a feature that is absent in the majority of other devices that fulfil the function of Boolean logic gates. Since both IMPLY and NAND functions are sufficient to make a circuit of arbitrary complexity, we concluded that singleton {NOT} and doubleton {YES; NOT} gates can act as functionally complete sets in DNA-integrated computational circuits. 

## 5. Conclusions

In conclusion, two DNA 4J gates with YES and NOT Boolean functions can be connected to make IMPLY, while two NOT gates can make a NAND function. Theoretically, a computational circuit of any complexity can be built only from this set of DNA logic gates. This opens a route to building computational circuits of arbitrary complexity from simple YES and NOT DNA logic gates. This modular connectivity could ease the burden of developing new architectures when realizing new Boolean circuitries. Therefore, while developing molecular logic gates, we should look for opportunities that are unexpected from our experience with electronic computers.

## 6. Patents

A provisional patent was filed for this technology.

## Figures and Tables

**Figure 1 nanomaterials-14-00600-f001:**
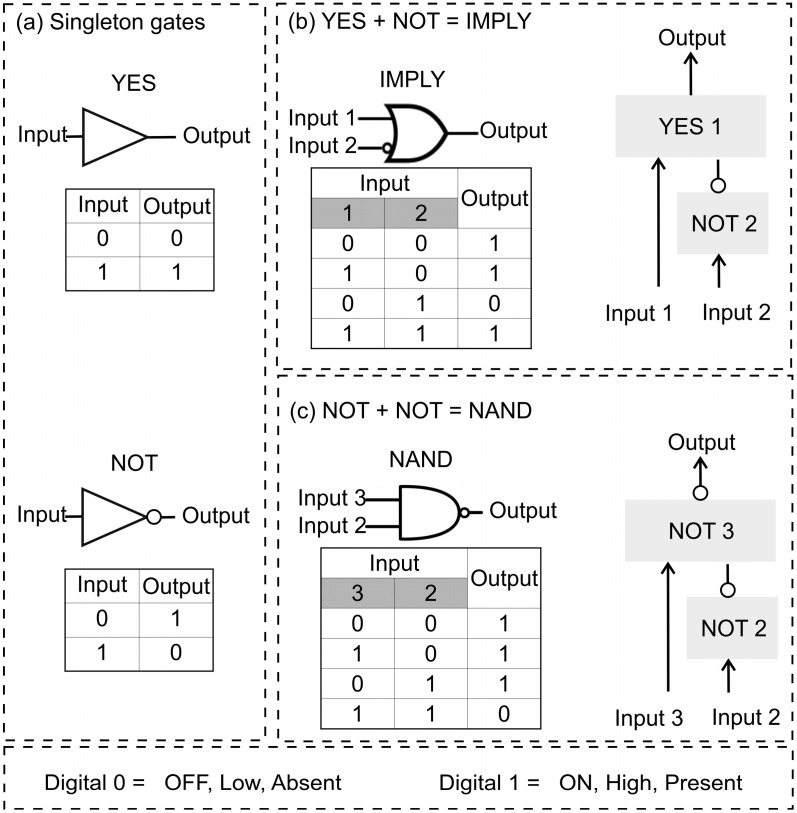
Boolean logic symbols and truth tables. (**a**) Singleton YES (top) and NOT (bottom) gates, (**b**) connecting YES 1 + NOT 2 to make IMPLY logic, (**c**) connectivity of two NOT gates (NOT 2 + NOT 3) to obtain a NAND logic function.

**Figure 2 nanomaterials-14-00600-f002:**
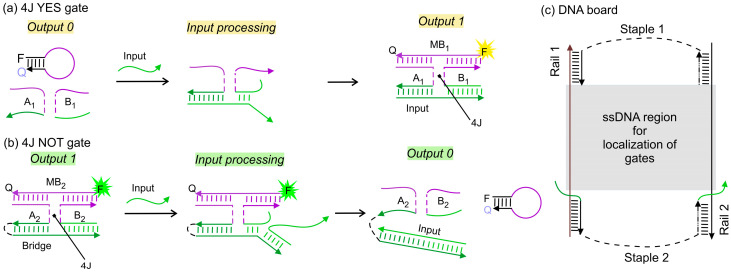
Components for functionally complete 4J gates. (**a**) The 4J YES gate before (left) and after (right) input recognition, digital output 0 and 1, respectively. (**b**) The 4J NOT gate in the absence (left) and presence (right) of the input; digital output 1 and 0, respectively. Labels Q and F in the MB strands represent a molecular quencher and a fluorophore, correspondingly. (**c**) DNA board. The grey shaded area represents the ssDNA region within the DNA board accessible for hybridization with the gate units. The dashed lines represent the oligoethylene glycol spacers (see Table 1 for details). The duplexes between rail fragments and complementary fragments of Staple 1, Staple 2, or the gate units are 10–11 base pairs, which correspond to one helical turn in B-DNA.

**Figure 3 nanomaterials-14-00600-f003:**
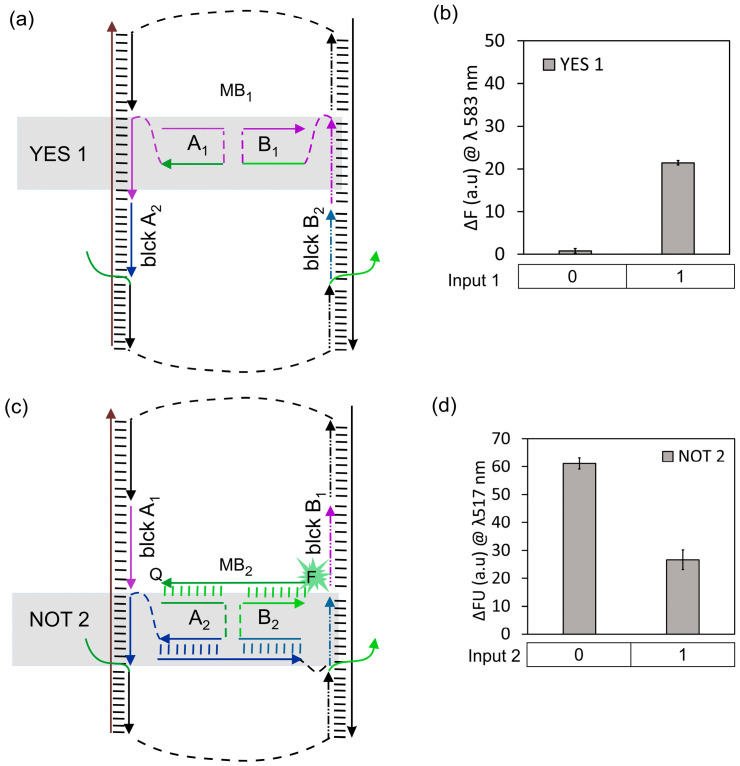
Functionally complete 4J gates integrated on DNA board. The 4J YES 1 (**a**) and NOT 2 gates (**c**) on the DNA board, both in the absence of input; ssDNA blocker strands (blck A_1_, A_2_, B_1_ and B_2_) fill the Rail fragments lacking the gates. Fluorescence response of 4J YES 1 (**b**) and 4J NOT 2 (**d**), respectively.

**Figure 4 nanomaterials-14-00600-f004:**
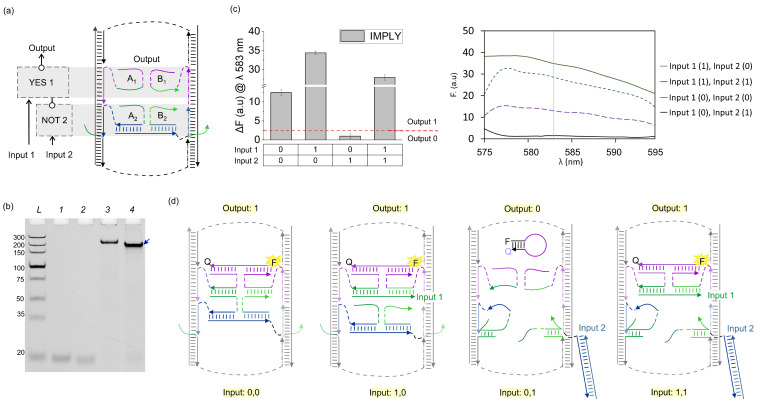
YES 1 + NOT 2 = IMPLY. (**a**) Localization and connectivity of YES 1 and NOT 2 on the DNA board. (**b**) The 8% native PAGE–50 mM MgCl_2_ results. L: dsDNA markers with their length, in base pairs, indicated to the left, 1: YES 1 gate strands (A_1_ + B_1_), 2: NOT 2 gate strands (A_2_ + B_2_ + Bridge), 3: DNA board only, 4: IMPLY full assembly (YES 1 + NOT 2 + DNA board). The blue arrow indicates the fully assembled IMPLY gate nanostructure. (**c**) Fluorescence of IMPLY upon excitation at 555 nm. Red dashed line represents an experimental threshold, which was calculated as the average fluorescence of YES 1’s output 0 plus three standard deviations. (**d**) Expected structural changes in the IMPLY nanostructure for the four Input 1/Input 2 combinations: digital inputs 0, 0; 1, 0; 0, 1; 1, 1.

**Figure 5 nanomaterials-14-00600-f005:**
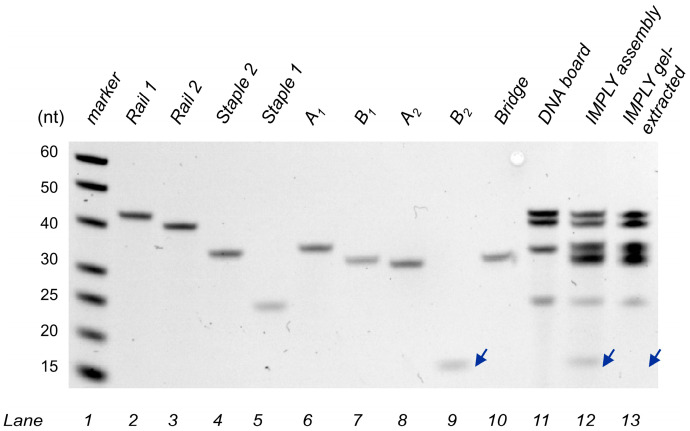
dPAGE analysis of full IMPLY assembly. The 12% dPAGE–8 M urea results. Lane 1: ssDNA markers with their lengths, in nucleotides, indicated; 2–10: individual ssDNA components of the IMPLY assembly; 11: DNA board; 12: IMPLY assembly before PAGE extraction. 13: IMPLY assembly after PAGE extraction. Blue arrowheads indicate the mobility of B_2_.

**Figure 6 nanomaterials-14-00600-f006:**
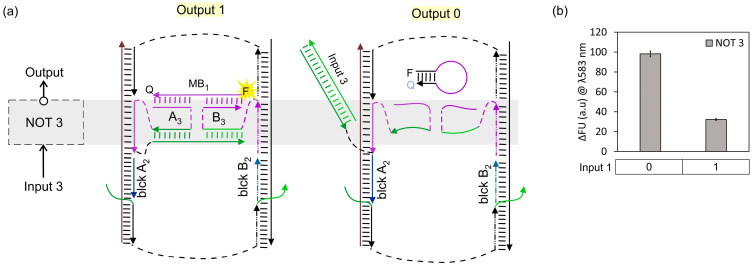
Individual response of NOT 3 on DNA board. (**a**) The 4J NOT 3 gate on DNA board; left: in the absence of input; right: in the presence of input; ssDNA blocker strands blck A_2_, and B_2_ hybridized to ssDNA board area lacking gates. (**b**) The 4J NOT 3 fluorescence response after exciting at λ: 555 nm.

**Figure 7 nanomaterials-14-00600-f007:**
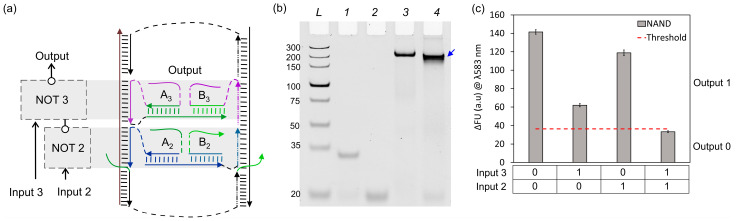
NOT 2 + NOT 3 = NAND. (**a**) Schematic representation of localization and connectivity of NOT 2 and NOT 3. (**b**) The 8% native PAGE–50 mM MgCl_2_ results. L: dsDNA markers with their length, in base pairs, indicated, 1: NOT 3 gate strands (A_3_ + B_3_), 2: NOT 2 gate strands (A_2_ + B_2_ + Bridge), 3: DNA board only, 4: NAND full assembly (NOT 2 + NOT 3 + DNA board). (**c**) Fluorescence response of NAND upon excitation at 555 nm. Red dashed line represents an experimental threshold, which was calculated as the average fluorescence of NAND’s output 0 plus three standard deviations.

**Figure 8 nanomaterials-14-00600-f008:**
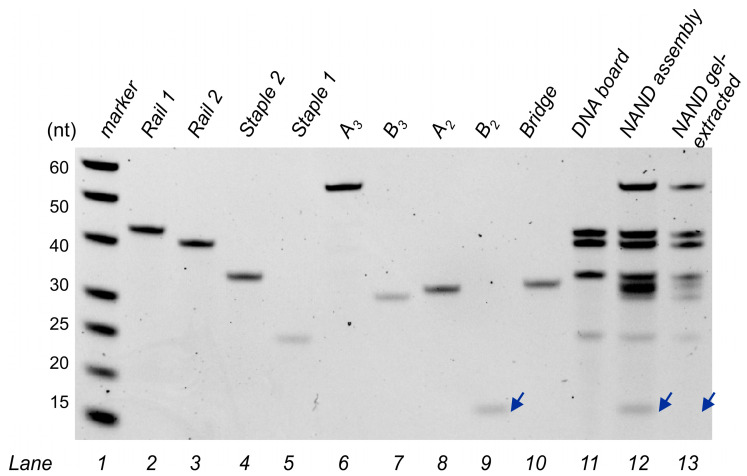
dPAGE analysis of the full NAND assembly. The 12% dPAGE–8 M urea results. Lane 1: ssDNA markers with their lengths, in nucleotides, specified; 2–10: individual ssDNA components of the NAND assembly; 11: DNA board; 12: NAND assembly before PAGE extraction. 13: NAND assembly after PAGE extraction. Blue arrowheads indicate the mobility of B_2_.

**Table 1 nanomaterials-14-00600-t001:** Oligonucleotides used in this study.

Name	Comments	Sequence
DNA Board		
Rail 1		CCT ATC GTG TT TTG TCG CTGA CCA TC GTA TCG CTT CGT CTATG
Rail 2		CTGAG TGAAT GAG CT CTA CA C TGC AGT ACC AC CGT TAG TCA
Staple 1		ATTCA CTCAG/iSp18//iSp18/CATAG ACG AAG
Staple 2		GACA AA CAC GAT AGG/iSp18//iSp18/TGA CTA ACG GT CCAG
Blck A_1_		CGA TAC GAT GG
Blck B_1_		TGT AGA GCTC
Blck A_2_		TCAG CGA CAA
Blck B_2_		GGT ACT GCA G
YES 1		
A_1_		*CT TTG TTC*/iSp18/**A GAC AAT GTA GC**/iSp18/CGATAC GATGG
B_1_		AGTAG AGCTC/iSp18/**GAAAC CCA GC**/iSp18/*GAT G ATT CC*
NOT 2		
A_2_		*TA CAT TGTC T*/iSp18/GGT GAAC C/iSp18/TCAG CGA CAA
B_2_		TG TTG CTC/iSp18/*GCT GGG*
Bridge		**AGGG GTT CAC CGA GCA ACA TTC**/iSp9/GGT ACT GCA G
NOT 3		
A_3_		*CT TTG TTC*/iSp18/A GAC AAT G/iSp18/CGATAC GATGG/iSp18/**GC TAC ATT GTCT GC TGG GTTTC**
B_3_		AGTAG AGCTC/iSp18/AAC CCA GC/iSp18/*GAT G ATT CC*
Inputs		
Input 3		GAAAC CCA GC AGAC AAT GTA GC
Input 2	hsa-miR-409-3p	/5′-Phos/-rGrArA rUrGrU rUrGrC rUrCrG rGrUrG rArArC rCrCrC rU
Input 1	hsa-miR-221-3p	/5′-Phos/rArGrC rUrArC rArUrU rGrUrC rUrGrC rUrGrG rGrUrUrUrC
Molecular Beacons (MB)		
MB_1_		/56-TAMN/CCT *GG AATCATC GAACAAAG* CA CAG CCAGG-3′-BHQ2
MB_2_		/56-FAM/CCAGG *CCCAGC AGACAATGTA* CCT GG/3BHQ_1/

Sequences of the same colour in different strands are complementary to each other; the colour code corresponds to that shown in Figures. Italic sequences indicate MB complementarity; underlined sequences, gate connectivity; bold sequences, input complementarity. Each sequence is entered as 5′->3′; iSp9 and iSp18 are oligoethylene glycol spacers 9 and 18 from IDT;/5Phos/, 5′ terminal phosphate group; r indicates ribonucleotide.

## Data Availability

All data are contained within the article.

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
