# Peer review of "Singleton {NOT} and Doubleton {YES; NOT} Gates Act as Functionally Complete Sets in DNA-Integrated Computational Circuits"

_nanomaterials, 2024, doi:10.3390/nano14070600_

Round 1
Reviewer 1 Report
Comments and Suggestions for Authors
In this research, they seem to connect YES (buffer) with NOT (inverter), and two NOT four-way Junction (4J) DNA gates to obtain IMPLY and NAND Boolean functions, respectively, each of which represents a functionally complete gate.
The research itself is valuable. However, the manuscript itself was not prepared in well-understandable ways. In the main text, design and expected operations are summarized. Only when I read this main text, this research looks like proposal of connected RNA research gates. However, when I read supporting information, lots of valid experimental data are presented. This presentation style of the current submission is not good. Only conceptual parts are published in the main text as Communication and really important data are separated into supporting information. This is not well understandable contribution. Instead of submitting this work as Communication, I recommend the authors to shift all the supporting information to the main text. The combined contents are submitted as a full paper. The authors have to logically integrate the main text and supporting information to make complete full paper.
Although this research aims to make logic circuit, the manuscript was not logically prepared. I recommend the authors resubmit this contributions after making combined manuscript.
Author Response
We moved all the results and discussion from the supporting information to the main manuscript. These changes are highlighted grey in the main manuscript.
Reviewer 2 Report
Comments and Suggestions for Authors
The manuscript of Bardales et al. entitled “Singleton {NOT} and Doubleton {YES; NOT} Act as Functionally Complete Sets in DNA Integrated Computational Circuits” describes the connection of YES/NOT and two NOT four-way Junction DNA gates affording IMPLY and NAND systems in a functionally complete gate. The manuscript is well presented, the figures are very descriptive and clear, and the authors made an interesting scientific contribution. It involves exploring the state of the art in the field of study. Was Figure 4c constructed from fluorescence spectra? If so, it would be interesting for the authors to include these spectra in the Supplementary material. I suggest the authors describe a little more about the potential applications of the elaborate device. I recommend evaluating the possibility of including figures from the supplementary material in the manuscript.
Author Response
Reviewer 2, per your suggestion we added the fluorescence spectra from which Figure 4c graph was constructed. Additionally, we also combined all the results and discussion from the supporting information with the main manuscript. These changes are highlighted in the main manuscript. Lastly, we provided insights on the potential applications in the conclusion section, line 271-278.
“In conclusion, two DNA 4J gates with YES and NOT Boolean functions can be connected to make IMPLY, while two NOT gates can make a NAND function. Theoretically, a computational circuit of any complexity can be built only from this set of DNA logic gates. This opens a route to building computational circuits of arbitrary complexity from simple YES and NOT DNA logic gates. This modular connectivity could ease the burden of developing new architectures when realizing new Boolean circuitries. Therefore, while developing molecular logic gates we should look for opportunities unexpected from our experience with electronic computers.”
Round 2
Reviewer 1 Report
Comments and Suggestions for Authors
Replies and revisions are fine. The revised version becomes acceptable.
Author Response
We thank Reviewer 1 for the positive comment and their time devoted to this manuscript.